# Antagonizing the S1P-S1P3 Axis as a Promising Anti-Angiogenic Strategy

**DOI:** 10.3390/metabo15030178

**Published:** 2025-03-05

**Authors:** Sofia Avnet, Emi Mizushima, Beatrice Severino, Maria Veronica Lipreri, Antonia Scognamiglio, Angela Corvino, Nicola Baldini, Margherita Cortini

**Affiliations:** 1Department of Biomedical and Neuromotor Sciences, University of Bologna, 40138 Bologna, Italy; sofia.avnet3@unibo.it; 2Department of Orthopaedic Surgery, School of Medicine, Sapporo Medical University, Sapporo 060-8543, Hokkaido, Japan; oomagari922emi@gmail.com; 3Department of Pharmacy, School of Medicine, University of Naples Federico II, 80131 Napoli, Italy; bseverin@unina.it (B.S.); antonia.scognamiglio@unina.it (A.S.); angela.corvino@unina.it (A.C.); 4Biomedical Science, Technologies, and Nanobiotechnology Lab, IRCCS Istituto Ortopedico Rizzoli, 40136 Bologna, Italy; mariaveronica.lipreri@ior.it

**Keywords:** sphingosine-1-phosphate, sphingosine-1-phosphate receptor 3, pepducins, tumor angiogenesis, osteosarcoma

## Abstract

Background: Angiogenesis, the process of new blood vessel formation, is critically regulated by a balance of pro- and anti-angiogenic factors. This process plays a central role in tumor progression and is modulated by tumor cells. Sphingosine-1-phosphate (S1P), a bioactive lipid signaling molecule acting via G-protein-coupled receptors (S1PR1–5), has emerged as a key mediator of vascular development and pathological angiogenesis in cancer. Consequently, targeting the S1P-S1PRs axis represents a promising strategy for antiangiogenic therapies. This study explores S1PR3 as a potential therapeutic target in osteosarcoma, the most common primary bone malignancy, which we have previously demonstrated to secrete S1P within the acidic tumor microenvironment. Methods: The effects of KRX-725-II and its derivatives, Tic-4-KRX-725-II and [D-Tic]4-KRX-725-II—pepducins acting as S1PR3 antagonists as allosteric modulators of GPCR activity—were tested on metastatic osteosarcoma cells (143B) for proliferation and migration inhibition. Anti-angiogenic activity was assessed using endothelial cells (HUVEC) through proliferation and tubulogenesis assays in 2D, alongside sprouting and migration analyses in a 3D passively perfused microfluidic chip. Results: S1PR3 inhibition did not alter osteosarcoma cell growth or migration. However, it impaired endothelial cell tubulogenesis up to 75% and sprouting up to 30% in respect to controls. Conventional 2D assays revealed reduced tubule nodes and length, while 3D microfluidic models demonstrated diminished sprouting area and maximum migration distance, indicating S1PR3’s role in driving endothelial cell differentiation. Conclusions: These findings highlight S1PR3 as a critical regulator of angiogenesis and posit its targeting as a novel anti-angiogenic strategy, particularly for aggressive, S1P-secreting tumors with pronounced metastatic potential and an acidic microenvironment.

## 1. Introduction

Angiogenesis, the process of forming new blood vessels, is a complex and dynamic mechanism regulated by a balance of pro- and anti-angiogenic factors. In addition to supporting tissue homeostasis under physiological conditions, it also plays a critical role in supporting tumor growth, invasion, and metastasis [1]. When solid tumors exceed a volume of 1–2 mm^3^, the surrounding tissue can no longer support their expansion, leading to a hypoxic, ischemic, acidic tumor microenvironment characterized by elevated interstitial pressure [2]. This hostile environment induces the release of growth factors and cytokines, promoting angiogenesis and lymphangiogenesis to satisfy the tumor’s metabolic demands. Unlike normal vasculature, tumor-associated blood vessels are irregular—they exhibit abnormal branching patterns, disorganized endothelial linings, discontinuous basement membranes, and inconsistent layers of smooth muscle and pericytes [3].

Lipid metabolism plays a key role in angiogenesis in both physiological and pathological environments [4]. De novo fatty acid synthesis supports endothelial cell proliferation, migration, and capillary and membrane formation [5,6], facilitated by carriers such as Fatty Acid Binding Protein 4 (FABP4) [7,8]. In cancer, lipids additionally contribute to energy storage, signaling, and protection from reactive oxygen species [9,10,11].

Among the lipids, sphingosine-1-phosphate (S1P) is a potent bioactive mediator with pleiotropic roles in cell migration, proliferation, survival, and differentiation in various cell types [12,13]. In endothelial cells, S1P regulates vascular developments and fibrosis [14] via G-protein-coupled receptors (S1PR_1–5_), with S1PR_3_ being the primary driver of angiogenesis [13]. Elevated S1P levels are also implicated in multiple cancers, including ovarian, prostate, colorectal, breast, and hepatocellular carcinoma [13], underscoring the potential of the S1P signaling pathway as a therapeutic target.

Our prior work revealed that osteosarcoma (OS) 143B cells secrete high S1P levels, and targeting the S1P-sphingomyelin pathway selectively impairs therapy-resistant, aggressive subpopulations within an acidic tumor niche, and it reduces xenograft tumor growth [15]. At the clinical level, serum S1P concentrations in OS patients decline following chemotherapy, suggesting a direct correlation between S1P and tumor progression [15]. These data, along with evidence from ovarian and lung cancers linking high S1P levels to metastatic spread [16,17,18,19] and poor prognosis [20,21], further highlight the therapeutic potential of S1P axis inhibition in OS. Targeting S1P-S1PR signaling could impair aggressive cancer cells while modulating angiogenesis, thereby suppressing vascularization and metastasis. However, the effect of S1PR inhibition on OS angiogenesis remains unexplored.

Here, we focused on selective S1PR_3_ antagonists, given the S1PR_3_ established role in angiogenesis [13,14,22] as a promising novel therapeutic strategy to disrupt OS angiogenesis. We evaluated three pepducins—allosteric S1PR_3_ modulators [14,23,24]: KRX-725-II (Myristoyl-GRPYDAN-NH_2_), which exhibits partial S1PR_1_/S1PR_3_ cross-reactivity, and its derivatives, Tic-4-KRX-725-II and [D-Tic]4-KRX-725-II, with enhanced S1PR_3_ selectivity [14]. To assess their efficacy, we combined conventional 2D in vitro assays of endothelial cell proliferation and tubulogenesis with an innovative 3D microfluidic model of endothelial cell sprouting within a passively perfused extracellular matrix. Notably, while the pepducins did not inhibit OS cell proliferation and migration, they critically reduced angiogenesis, demonstrating for the first time the potential of S1PR_3_ as an antiangiogenic target in OS. Overall, this study provides a rationale for integrating S1P-targeting strategies into multi-modal OS treatments to limit tumor progression and metastasis.

## 2. Results

### 2.1. S1P Receptor Expression in Endothelial Cells and Osteosarcoma Cells

To elucidate the effectiveness of the sphingomyelin-S1P pathway as a therapeutic target in anti-angiogenic therapies in OS, we first reasoned to assess the expression of the five different S1P receptors in HUVEC (endothelial cells) versus 143B OS cells. As shown in Figure 1, HUVEC and 143B cells express comparable S1PR_3_ levels. In contrast, relative to HUVECs, 143B cells display notably higher levels of S1PR_2_ and S1PR_5_ receptor expression and significantly lower expression of S1PR_1_.

### 2.2. Osteosarcoma 143B Cells Are Unaffected by the Inhibition of S1P3 Receptor

Given the critical role of the sphingomyelin-S1P pathway in 143B cell survival and migration [15], we aimed to determine whether S1P signals through the S1PR_3_ in OS cells. We treated a monolayer of 143B cells with three concentrations (10–30–100 μM) of pepducins to verify their effects on proliferation. As shown in Figure 2A–C, except for the highest dose of KRX-725-II, the compounds did not affect cell number, suggesting minimal S1PR_3_ involvement in OS cell proliferation.

Since previous studies have shown consistent differences between 2D and 3D models in OS [15,25], we further evaluated the effects on 143B cells grown in 3D using an indirect viability assay (Alamar Blue assay, Figure 2D). Again, S1PR_3_ inhibition did not reduce cell viability, suggesting that the shift from 2D to 3D culture does not modify S1PR_3_ activation or pro-viability and pro-proliferation function.

To assess tumor cell migration, 143B cells were grown as 3D spheroids and then allowed to adhere and migrate in a flat-bottomed 96-well plate in the presence of the drugs. Migration was quantified as the difference in spheroid spread area between the initial time point (0 h, black circles) and 24 h post-treatment (red circles), a metric reflecting invasive potential akin to in vivo tumor behavior. Unexpectedly, drug-treated spheroids showed a trend toward increased migration, though no treatment significantly impaired migration compared to controls (Figure 2E,F). Overall, these findings suggest S1PR_3_ plays a negligible role in 143B OS cell migration.

### 2.3. Inhibition of S1P3 Receptor Does Not Affect Endothelial Cell Proliferation

Next, we evaluated the role of the S1PR_3_ in endothelial cell proliferation. HUVEC cells were cultured as a 2D monolayer and treated with S1P3 receptor antagonists (10–30–100 μM) (Figure 3A). Cell growth was monitored for up to 72 h post-treatment. None of the three compounds had any significant effect on cell proliferation, even at the highest concentrations (Figure 3A). In line with this, we tested the effect on HUVEC cells cultured as 3D spheroids (Figure 3B); again, the pepducins did not reduce endothelial cell viability.

### 2.4. Inhibition of S1P3 Receptor Profoundly Impairs Endothelial Cell Tubulogenesis

Physiologically, endothelial cells must proliferate, migrate toward proangiogenic growth factor stimuli, and further commit and differentiate, forming lumen-containing tubes (tubulogenesis) to facilitate blood flow in later stages of angiogenesis. Here, we aimed to evaluate the effects of S1PR_3_ antagonists on tubulogenesis. HUVEC cells were seeded on a Matrigel^®^ layer (Figure 4A) and imaged via time-lapse over 24 h to monitor tubule formation. The number of nodes and tubule length were quantified at 12 h post-seeding, when maximum length occurred just before tubule shrinking and apoptosis. As shown in representative images (Figure 4B) and quantified (Figure 4C), all compounds significantly reduced the total length of capillary tubes, even at the lowest concentration. Additionally, a significant decrease in the average branch points was revealed with anti-S1PR_3_ antagonists (Figure 4C, lower graph). In more detail, for the reduction in total length and number of branch points, we calculated an IC50 of 19.7 μM, 25.9 μM, and 24.5 μM for the compounds with IC50 of the three compounds KRX-725-II, Tic-4-KRX-725-II, and [D-Tic]4-KRX-725-II, respectively.

These results indicate that the S1P3 receptor has an important role in the differentiation process that drives endothelial cell differentiation, with particular regard to tubulogenesis.

### 2.5. Angiogenesis Is Deeply Impaired by S1P3 Receptor Modulation

To better replicate the in vivo microenvironmental conditions, which are present both in physiological and pathological contexts, such as cancer, we performed vessel formation studies using 3-lane microfluidic chambers (Figure 5A).

We injected Matrigel^®^ into the ECM gel channel of a microfluidic chamber, followed by seeding GFP-positive HUVEC cells in the top perfusion channel. The vessels were then exposed to an angiogenic cocktail of the well-established angiogenic factors S1P and VEGF [26,27], and the chamber underwent passive perfusion with endothelial cell medium (Figure 5A). Vessel formation was assessed using confocal microscopy after fixing and staining the vessels for VE-cadherin, an intercellular junction marker. As shown (Figure 5B), side and top views revealed the formation of 3D capillary-resembling structure formation.

Next, we tested the effect of S1PR_3_ antagonists on live vessel sprouting. After vessel formation, the tested compounds were added (Figure 5C). Sprouting through the ECM was measured on live cells 24 h after the addition of the tested compounds via maximum intensity projection imaging, excluding cells that migrated on top of the phase guide (Figure 5D). We also measured the maximum cell migration distance within the ECM (Figure 5E). Interestingly, all inhibitors significantly reduced endothelial sprouting and migration at a concentration of 100 μM, but not at 10 μM, further emphasizing the S1P-S1PR_3_ axis as a potential target for antiangiogenic therapies in S1P-secreting cancers.

## 3. Discussion

This study explored the role of the S1P-S1PR_3_ axis in tumor and endothelial cell function to unveil its therapeutic relevance in anti-angiogenic strategies to treat OS. While 143B OS cells expressed distinct S1P receptor profiles compared to HUVECs, S1PR_3_ inhibition showed negligible effects on OS cell proliferation, viability, or migration. Similarly, endothelial cell proliferation remained unaffected by S1PR_3_ antagonists. However, S1PR3 blockade significantly impaired tubulogenesis, vessel sprouting, and endothelial migration.

Angiogenesis is essential for tumor growth and metastasis, making it a critical focus in cancer therapy [28,29,30]. Current anti-angiogenic approaches predominantly target VEGF, such as bevacizumab, the first VEGF inhibitor approved for clinical use [31]. However, anti-VEGF monotherapies frequently yield limited durability due to resistance mechanisms and tumor adaptation [32]. Prolonged anti-angiogenic treatments risk excessive vessel pruning, hypoxia, and compensatory proangiogenic pathways [33,34,35], necessitating strategies that target multiple mechanisms. Combination therapies integrating antiangiogenics with complementary agents have shown potential in advanced OS, where VEGF-centric strategies have not delivered the expected outcomes [36].

One emerging area of interest lies in targeting the S1P signaling axis, a key mediator of physiological and pathological angiogenesis. While drug development has prioritized S1P1 receptors—spurred by the non-specific S1P receptor agonist fingolimod (approved by the FDA in 2010 and EMA in 2011 for multiple sclerosis [37])—S1PR_3_’s established angiogenic role [13] positions it as a compelling candidate for therapeutic exploitation.

To explore the role of S1PR_3_ in angiogenesis, we investigated pepducin-based antagonists, including the first-generation compound KRX-725-II (which exhibits partial S1PR_1_/S1PR_3_ cross-reactivity [24]), and its derivatives, Tic-4-KRX-725-II and [D-Tic]4-KRX-725-II. While our prior work demonstrated that inhibiting the S1P-sphingomyelin pathway compromises OS cell migration and survival [15], our current findings suggest that this effect is not mediated by S1PR_3_. In 143B cells, which express higher S1PR_2_ and S1PR_5_ levels than HUVECs, compensatory signaling via these receptors—particularly S1PR_2_, which shares Gi pathway activation with S1PR_3_ and S1PR_1_—may offset S1PR_3_ blockade [22]. However, we do not exclude that, in OS, S1PR_3_‘s role may be more critical in cancer stem cells, which drive aggressive tumor phenotypes [38]. The resistance observed in 143B cells to the tested pepducines may also stem from tumor-specific lysosomal adaptations. Previous studies reported that certain sarcomas develop drug resistance through elevated vacuolar ATPase (V-ATPase) expression and activity, increasing lysosomal acidity. As a demonstration, targeting V-ATPase with the proton pump inhibitor omeprazole-sensitized 143B OS cells to doxorubicin in a xenograft model [39], suggesting that the high number of acidic lysosomes in these cells contributes to drug resistance. Furthermore, extracellular acidosis, a hallmark of tumor progression [15], exacerbates both lysosome number and acidity [39]. Our hypothesis is that in spheroid cultures, localized acidosis and ECM secretion [25] that form in the inner spheroid regions likely further reduced pepducine penetration and effectiveness, as previously observed with doxorubicin.

In contrast, S1PR_3_ inhibition in HUVECs did not affect proliferation but significantly impaired tubule formation during differentiation, underscoring S1PR_3_’s preferential role in vascular maturation over mitotic activity. Notably, all three compounds exhibited comparable effects in proliferation or tubulogenesis assays, reinforcing the S1PR_3_ dominance over S1PR_1_ in endothelial cell differentiation. However, in a microfluidic 3D sprouting model, KRX-725-II more potently inhibited HUVECs sprouting, suggesting that the S1PR_1_ contributes to this process. These findings indicate that endothelial cells adapt their receptor expression profiles as they progress through different stages of commitment. Supporting this, tumor endothelial S1PR_1_ overexpression normalizes vasculature while influencing tumor growth and metastasis in lung cancer [40], highlighting context-dependent receptor roles across tumor types.

Traditional 2D in vitro experiments fail to fully replicate the in vivo microenvironments experienced by endothelial cells, which are exposed to hemodynamic forces like shear stress and physiological oxygen levels (typically <10%) rather than ambient conditions (21%). To address these limitations, microfluidic technologies offer physiologically relevant platforms that better emulate vascular dynamics [41,42,43]. Here, we employed a microfluidic device where endothelial cells in the upper channel formed a vessel wall-like structure against the Matrigel^®^, which was injected into the middle channel, while an angiogenic cocktail in the lower channel stimulated cell sprouting. This setup enabled passive perfusion, mimicking shear stress, and the formation of gradients of nutrients and their exchange within a 3D ECM (Matrigel). In these conditions, pepducins inhibited vessel sprouting only at 100 μM. This discrepancy in respect to results obtained with conventional tubulogenesis assays—where endothelial cells form simplified 2D networks on a Matrigel^®^ layer—likely stems from the self-organization ability of endothelial cells to recapitulate 3D vascular complexity that is allowed and facilitated in such devices, more closely mimicking physiological conditions. Such physiological fidelity can alter drug responsiveness, underscoring the need for empirical dose optimization in 3D systems, as observed in other cancer models [44].

Our findings validate microfluidics as a robust in vitro tool for the study of angiogenesis [2,45,46] and reinforce the role of S1PR_3_ as a therapeutic target in anti-angiogenic strategies. Specifically, targeting S1PR3 may disrupt tumor-supporting vascular branching, offering a novel approach to impair cancer progression.

This study represents a preliminary investigation, and further research is necessary to determine physiologically relevant drug concentrations. Modifications such as nanocarrier delivery could enhance specificity toward tumor endothelial cells while minimizing off-target effects. Current clinical trials investigating S1P receptor modulators (www.clinicaltrial.gov (accessed on 11 February 2025)) primarily target conditions like multiple sclerosis and systemic lupus erythematosus, employing daily doses of 0.5 mg–20 mg, lower than our in vitro concentrations. However, dosing regimens are disease- and compound-specific, necessitating tailored optimization based on receptor targets, clinical context, and therapeutic goals.

Our results provide evidence for the role of S1PR_3_ in angiogenesis, yet limitations warrant consideration. First, using a single OS cell line (143B) necessitates validation across diverse OS models or even other cancer histotypes. Second, while 2D/3D in vitro models were employed, in vivo studies are essential to confirm therapeutic. Third, the downstream signaling mechanisms driving S1PR3-mediated endothelial effects remain uncharacterized. Finally, compensatory roles of other S1P receptors in OS angiogenesis require exploration.

Despite these limitations, our work strongly supports targeting the S1P-S1PR_3_ axis in anti-angiogenic strategies for OS and other S1P-secreting tumors.

In conclusion, based on our findings, S1PR_3_ emerges as a critical regulator of endothelial cell differentiation and vascular morphogenesis, with inhibition exerting more pronounced effects on endothelial versus tumor cells. Considering the critical role of angiogenesis in tumor progression and the need for multi-target therapies to overcome drug resistance, S1PR3 antagonism via pepducins presents a promising strategy, particularly in S1P-driven malignancies like OS.

## 4. Materials and Methods

### 4.1. Cells and Reagents

The KRX-725-II (Myristoyl-GRPYDAN-NH2) compound and its derivatives Tic-4-KRX-725-II and [D-Tic]4-KRX-725-II were synthesized as previously described [14].

HUVEC-GFP cells were purchased from Promocell (#C-12200) (Heidelberg, Germany), whereas 143B (#CRL-8303) was purchased from the American Type Culture Collection (ATCC, Washington, DC, USA). Cells were maintained at 37 °C in a humidified atmosphere with 5% CO_2_. All cell lines were tested against mycoplasma with nuclear staining every month. 143B cells were validated (LGC Standards, Milan, Italy) in 2023. HUVEC cells were grown to confluence in Endothelial Cell Growth Medium (Promocell) plus Endothelial Cell Growth Supplement (Promocell) in tissue culture flasks precoated with 0.2% gelatine in water (Merck, Darmstadt, Germany). HUVEC cells were used up to passage 8. 143B were cultured in IMDM (Life Technologies, Carlsbad, CA, USA) plus penicillin (20 U/mL), streptomycin (100 mg/mL), and 10% heat-inactivated fetal bovine serum for a range of 10–20 passages from thawing. All cells were maintained at 37 °C in a humidified atmosphere with 5% CO_2_.

For 3D models, to form spheroids, cells were grown in ultralow attachment 96-well plates (SBio, Hudson, NH, USA), 5 × 10^3^ cells/well in 200 μL medium/well; spheroids were treated with 100 μM pepducins and let grow for 72 h.

### 4.2. Primer Design, RNA Extraction, and Gene Expression

To assess the expression of S1PRs, primers were designed for each gene. Sequences were obtained on PubMed gene datasets (S1PR_1_: NM_001400.5; S1PR_2_: NM_004230.4; S1PR_3_: NM_005226.4; S1PR_4_: NM_003775.4; and S1PR_5_: NM_030760.5). Primers were designed with Primer3 software v. 0.4.0., and sequences were run on BLAST: Basic Local Alignment Search Tool and UCSC Genome Browser Home to assess for non-specific alignments. GC content, melting temperature, and self-dimer check were evaluated with Oligo Analysis Tool from Eurofingenomics.

RNA was extracted with Trizol reagent (ThermoFisher, Waltham, MA, USA). The total RNA was reverse transcribed into cDNA using RNase inhibitor and MuLV Reverse Transcriptase (Applied Biosystems, Foster City, CA, USA). First-strand cDNA was synthesized with RT-qPCR using random hexamers. Real-time PCR was performed by amplifying 500 ng using the SsoAdvanced Sybr Green Mix (Biorad, Hercules, CA, USA) and the CFX96Touch instrument (Biorad). Normalization was performed on three different housekeeping genes that were used, i.e., Gusb, YWHZ, and GAPDH. Relative gene expression was thus obtained with the ratio between the geometric mean of the expression of the housekeeping genes with the following sequences: GAPDH For: ccaaggagtaagacccctgg; GAPDH Rev: aggggagattcagtgtggtg; Gusb For: cccactcagtagccaagtca; Gusb Rev: gttctgctgctgtggaagtc; YWHZ For: ccgcatgatctttctggctc; YWHZ Rev: tagtctgtgggatgcaagca; S1PR1 For: ccaagaaattccaccgaccc; S1PR1 Rev: ccccagacaagagcaggtta; S1PR2 For: ggagtacctgaaccccaaca; S1PR2 Rev: cgcaacagaggatgacgatg; S1PR3 For: gtgctcggccagttacaaaa; S1PR3 Rev: tgacagcgagggtttgtttg; S1PR4 For: cgcttctgtgtgattctggg; S1PR4 Rev: tcgaacttcaatgttgccagg; S1PR5 For: gaggactcaggctaaggtgg; S1PR5 Rev: tgattcggaggggtcttcag.

### 4.3. Proliferation Assay

HUVEC cells were seeded in 96-well, flat-bottom tissue culture plates and coated with 0.2% gelatin. After seeding, cells were treated with 10 μM, 30 μM, or 100 μM of KRX-725-II, Tic-4-KRX-725-II, and [D-Tic]4-KRX-725-II, and cell growth was assessed up to 72 h after drug exposure. To obtain the total number, cells were stained in complete cell medium with Hoechst 33,342 2.25 μg/mL for 20 min. Images were acquired by fluorescent microscopy using the ImageXpress Pico Automated Cell Imaging System (Molecular Device, San Jose, CA, USA) with a 4× objective. Analysis software detected the total number of cells based on Hoechst-positive nuclei.

### 4.4. Alamar Viability Assay

To measure cell viability based on metabolic activity, spheroids were generated using ultralow attachment 96-well plates, as previously described [4]. Briefly, cells were grown in ultralow attachment 96-well plates (SBio), 5 × 10^3^ cells/well in 200 μL medium/well. Furthermore, 24 h after seeding, spheroids were treated with pepducins 100 μM, and after 72 h of cultures, medium was removed, and the spheroids were washed with PBS. Then, 100 µL of Alamar Blue solution (Thermo Fisher Scientific, Waltham, MA, USA) was added to each well and incubated for 6 h at 37 °C and 5% CO_2_, protected from light. Upon entering living cells, resazurin (the active molecule of Alamar blue) is reduced to resorufin in metabolically active cells. After incubation, the reaction solution was collected and moved to a new 96-well plate and measured with Tecan at 580 nm (Infinite F200pro Tecan, Männedorf, Switzerland).

### 4.5. Spheroid Migration Assay

143B cells were seeded into ultra-low attachment plates (SBio) at a density of 5 × 10^3^ cells per well in 200 µL of medium per well and treated with 100 µM pepducins. After 24 h, the spheroids were transferred to a flat-bottomed 96-well plate to allow for adhesion. Time-lapse imaging was conducted for an additional 24 h. The migration area was calculated by subtracting the spheroid area at the start of the time-lapse (black circles in Figure 2E) from the area measured after 24 h of adhesion (red circles in Figure 2E). Acquisition was obtained with the Xpress Pico microscope with a 4× objective in brightfield. The migration area was quantified using the Image J software v. 1.51j8.

### 4.6. Tubulogenesis

HUVEC cells were seeded in 96-well, flat-bottom tissue culture plates coated with 60 μL/w of 75% matrigel growth factor reduced (BD Biosciences, Erembodegem, Belgium) at a density of 2 × 10^4^ cells/well. Cells were then incubated with 10 μM or 100 μM pepducins, and tubulogenesis was imaged with time-lapse microscopy for 24 h, with images acquired every hour. Acquisition was obtained with the Xpress Pico microscope with a 4× objective; green (GFP) and brightfield emissions were measured. Capillary tube-like length and number of nodes were quantified using the Image J software.

### 4.7. Sprouting Assay

A single-cell suspension of HUVEC-GFP cells (2 × 10^4^ cells/channel) was obtained from a 70% confluent flask, detached in trypsin, and counted. Before cell seeding, a solution of 60% Matrigel^®^ (Corning, New York, NY, USA), 30% 30 mg/mL neutralized Rat Tail Collagen I (Thermo Fisher Scientific), and 10% endothelial cell culture medium was seeded in the middle channel of a three-lane device (Mimetas, Oegstgeest, The Netherlands).

Three-lane Mimetas^®^ microchambers were used for HUVEC vessel formation; cells were then seeded in the upper channel and Matrigel^®^ in the middle channel; vessels were allowed to form. After 24 h, vessel formation was assessed by GFP fluorescence on live cells with confocal microscopy, and an angiogenic cocktail formed of 250 nM S1P and 37.5 ng/mL of VEGF was added in the lower channel; 100 μM of pepducins were added in the top channel. To simulate physiological perfusion flow, the microfluidic device was placed on a rocking plate: the rocker was set to change the tilting angle every 8 min. Furthermore, 24 h after drug treatment, vessels were live imaged with GFP fluorescence.

For image acquisition of the middle channel, we used the A1R MP confocal microscope (objective 20× air, numerical aperture 0.75, refractive index 1, resonant scanning, zoom at 1, z-step 3 μm; the total Z-stack was 180 μm with Ni-E ZDrive, Nikon, Minato, Japan). Images show the maximum intensity projection of the whole Z-stack. For the analysis, we used NIS Elements AR 5.40.01.

### 4.8. Immunofluorescence

To perform immunostaining of HUVEC endothelial vessels, all the solutions were added to the inlets and outlets of a microfluidic three-lane device, and the volumes were adjusted according to the manufacturer’s instructions (Mimetas). The cells were washed once with PBS, fixed with 100% ice-cold methanol, and blocked with 1% BSA. Moreover, 100 μL were added to the top left inlet, and 50 μL were added in all other inlets. As primary antibodies, we used anti-VE cadherin (#33168, Abcam, Cambridge, UK), followed by incubation with anti-rabbit secondary TRITC 568 (1:500, Life Technologies). Antibodies were incubated in 25 μL and added to the top left and top right inlets. Nuclei were counterstained with Hoechst 33,258 (0.125 μg/mL, Thermo Fisher Scientific) in 25 μL, added to the top left and top right inlets.

### 4.9. Statistics

Statistical analysis was performed using the Graph Pad Prism 7.04 software for Windows (Graph Pad Software, La Jolla, CA, USA). Results were reported as mean ± standard error of the mean, and the differences among groups were analyzed using the nonparametric Mann–Whitney test for the difference between groups. Only *p* < 0.05 was considered significant.

## Figures and Tables

**Figure 1 metabolites-15-00178-f001:**
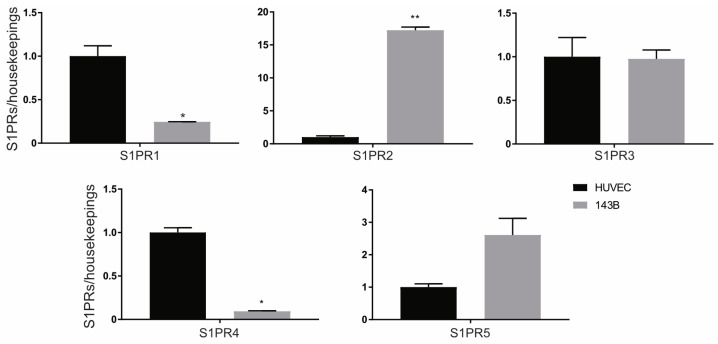
S1P receptor expression in HUVEC and 143B cells. Real-time PCR analysis of the indicated genes normalized to three housekeeping genes (GAPDH, GUSB, and YWAHZ) in 143B and HUVEC cells cultured as monolayers for 72 h. Data presented as mean ± S.E.M. Unpaired two-tailed Mann–Whitney U test (* *p* < 0.05 and ** *p* < 0.01 versus HUVEC; *n* = 3).

**Figure 2 metabolites-15-00178-f002:**
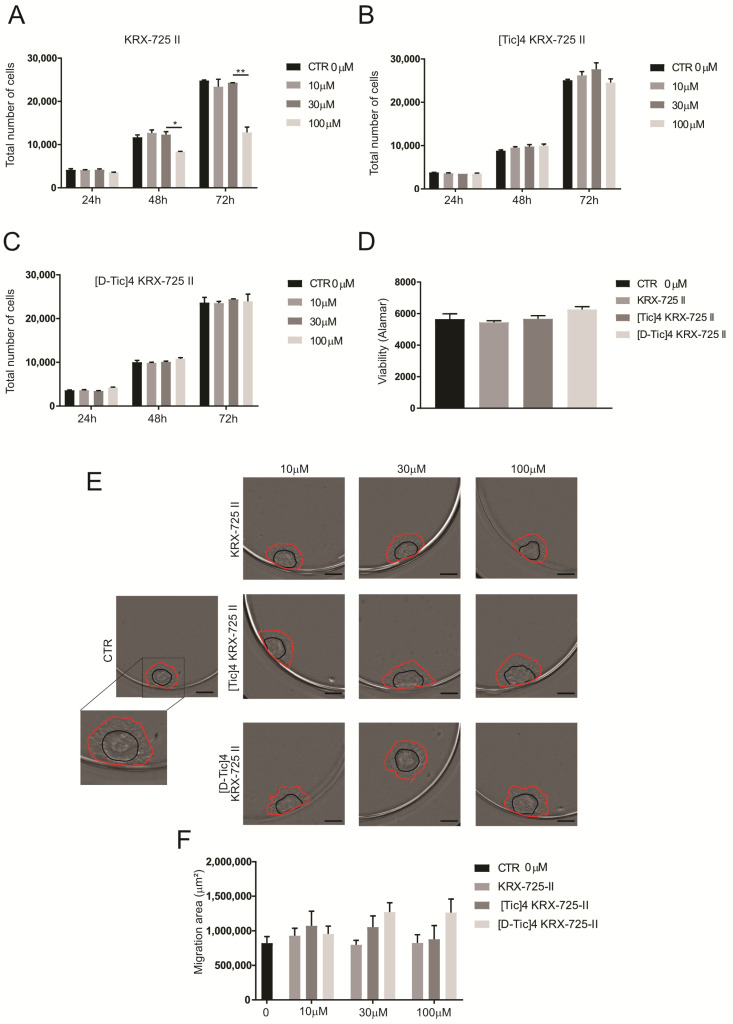
Inhibition of S1P3 does not affect OS cell proliferation or migration. (**A**–**C**) 143B OS cells were cultured in monolayer, incubated with the indicated compounds for 24, 48, or 72 h at the indicated concentrations. Total number of cells was assessed by staining of cell nuclei by Hoechst data presented as mean ± S.E.M. (unpaired two-tailed Mann–Whitney U test; * *p* < 0.05 and ** *p* < 0.01 versus control, *n* = 6). (**D**) Viability of OS spheroids (expressed as 580 nm absorbance). 143B were cultured as spheroids, treated with the indicated compounds at a concentration of 100 μM. The Alamar Blue assay was performed 72 h after the treatment (*n* = 12). (**E**) Migration of 143B OS spheroids as indicated, representative images. Spheroids formed in ultra-low attachment plates, moved to flat-bottom plates to allow adhesion, treated as indicated and imaged for 24 h (scale bar: 500 μm). (**F**) Quantification of the representative images shown in E. To measure the migration area, the area of the spheroids was assessed at time point 0 (black circle) and after 24 h (red circle). The graph shows the difference between the two areas (μm^2^) (*n* = 6).

**Figure 3 metabolites-15-00178-f003:**
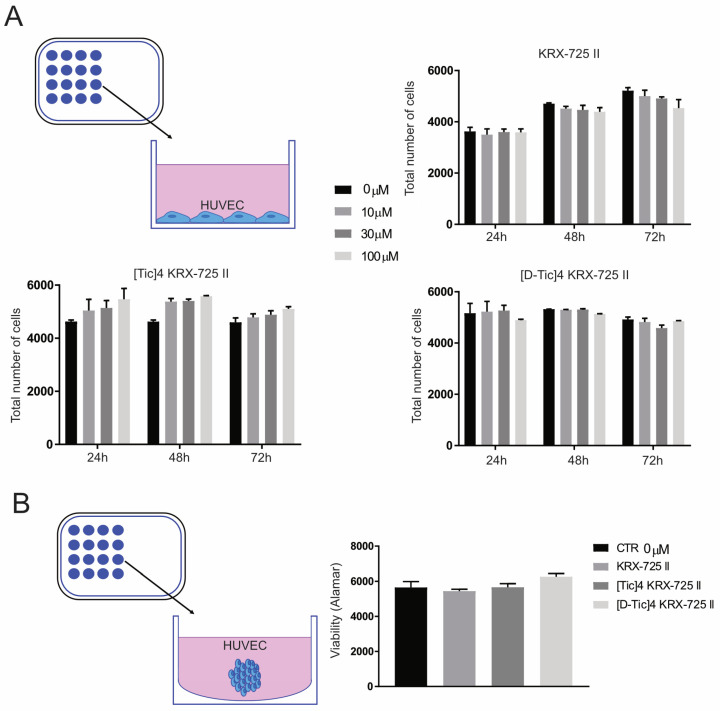
Pepducins do not affect endothelial cell proliferation. (**A**) Schematic representation of cells seeded in monolayer and graphs showing the results of quantification. HUVEC endothelial cells were cultured and incubated with the inhibitors for 24, 48, or 72 h at different concentrations. The total number of cells was assessed by staining of cell nuclei by Hoechst (*n* = 6). (**B**) Schematic representation of cells seeded as 3D spheroids in ultra-low attachment plates and graph related to the viability assay of HUVEC spheroid cells (expressed as 580 nm absorbance). HUVEC were treated with the inhibitors at a concentration of 100 μM. The Alamar Blue assay was performed 72 h after the treatment. (*n* = 12, data presented as mean ± S.E.M).

**Figure 4 metabolites-15-00178-f004:**
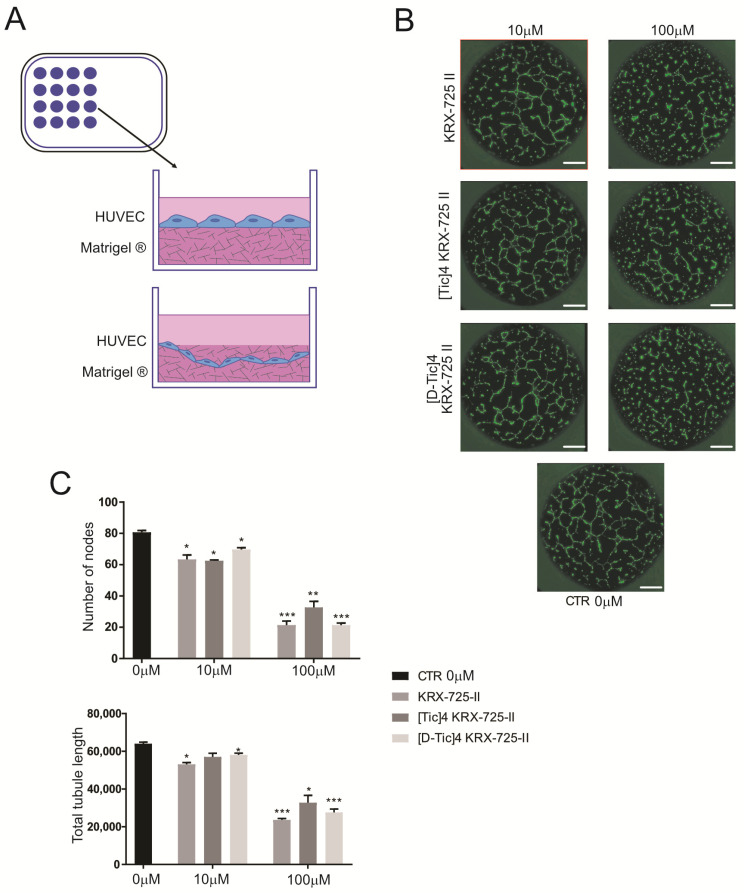
Tubulogenesis is impaired by S1PR_3_ inhibition. Data presented as mean ± S.E.M. (**A**) Schematic representation of the tubulogenesis experiment. HUVEC were seeded on top of a layer of 75% Matrigel^®^, and tubulogenesis was followed for 24 h post-seeding. (**B**) Representative images of the experiment schematized in (**A**). HUVEC-GFP cells were treated as indicated, and tubules were imaged 12 h post-seeding. Images show an overlay of brightfield and GFP acquisitions (scale bar: 500 μm). (**C**) Quantification of the number of nodes and total tubule length of the experiment shown in panel B (unpaired two-tailed Mann–Whitney test; * *p* < 0.05 versus control; ** *p* < 0.01 and *** *p* < 0.001 versus control, *n* = 6).

**Figure 5 metabolites-15-00178-f005:**
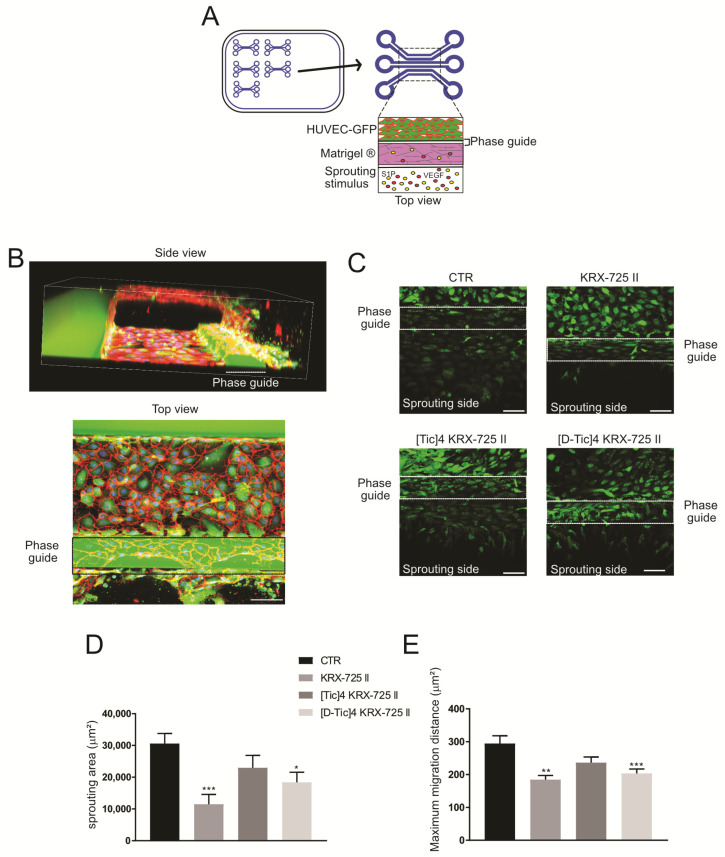
S1PR_3_ modulation decreases endothelial cell sprouting. (**A**) Schematic representation of vessel formation in a microfluidic platform. Three-lane Mimetas^®^ microchambers are used for HUVEC vessel formation; cells are seeded in the top channel, Matrigel^®^ is injected in the middle channel, and the lower channel is used for the addition of the angiogenic cocktail (S1P 250 nM and VEGF 37.5 ng/mL). HUVEC-GFP cells were allowed to form vessels for 24 h before the addition of the angiogenic cocktail or pepducins at a concentration of 100 μM (in the upper channel); 24 h after drug treatment, vessels were live imaged with GFP fluorescence. (**B**) Representative images showing 3D endothelial vessel formation from side and top view. HUVEC-GFP cells were fixed and stained with anti-VE cadherin (red) and Hoechst (nuclei, blue, scale bar: 100 μm). (**C**) HUVEC-GFP endothelial cells live sprouting from the upper channel to the Matrigel^®^ matrix, top view, representative images. The endothelial vessel was acquired on a 180 μm Z-stack, with images every 3 μm. Images show the maximum intensity projection of the whole Z-stack (scale bar: 100 μm). (**D**,**E**) Quantification of the sprouting area and maximum migration distance of the images shown in (**C**). Sprouting was measured and quantified using maximum intensity projection images, excluding cells on top of the phase guide, whereas the maximum migration distance was measured using maximum intensity projection images (unpaired two-tailed Mann–Whitney test; * *p* < 0.05, ** *p* < 0.01, and *** *p* < 0.001 versus control, *n* = 10).

## Data Availability

The raw data supporting the conclusions of this article will be made available by the authors on request.

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
