# Peer review of "Antagonizing the S1P-S1P3 Axis as a Promising Anti-Angiogenic Strategy"

_metabolites, 2025, doi:10.3390/metabo15030178_

Round 1
Reviewer 1 Report
Comments and Suggestions for Authors
For Authors:
In my opinion, the manuscript is well-designed and provides very good analysis, so it’s acceptable for publication after some minor revisions.
- In the abstract section (line 34, for example), please add some quantitative data.
- Please extend paragraph 3 (line 72) of the introduction and provide more information.
- Please improve the last paragraph of the introduction by focusing on novelty. It is better to move the presented details of your previous study to a separate paragraph before the last paragraph.
- Please provide more discussion about your approach findings in line 309.
Good luck!
Author Response
In my opinion, the manuscript is well-designed and provides very good analysis, so it’s acceptable for publication after some minor revisions.
We thank the reviewer for the nice comments about our manuscript, which has hopefully now been improved with his/hers suggestions.
- In the abstract section (line 34, for example), please add some quantitative data.
Response: We have now added the % of inhibition of the pepducins on tubulogenesis and sprouting.
2. Please extend paragraph 3 (line 72) of the introduction and provide more information.
Response: We have expanded paragraph 3 of the introduction, which now provides additional information about the clinical relevance of targeting the S1P axis. Lines: 71-81
3.Please improve the last paragraph of the introduction by focusing on novelty. It is better to move the presented details of your previous study to a separate paragraph before the last paragraph.
Response: We have separated the two paragraphs, as suggested, and expanded the part on the results to further enlighten the novelty. Lines: 82-94
4. Please provide more discussion about your approach findings in line 309.
Response: We have provided few more references that show the relevance of microfluidics in the study of angiogenesis and add additional details and explanation on the advantages of the microfluidic models. Lines: 319-335
Good luck!
Thank you!
Reviewer 2 Report
Comments and Suggestions for Authors
The following corrections/clarifications are required. The comments are being provided section-wise:
The introduction section is inadequate. The hypothesis is missing, and the aim is not well written. The structure of original articles should be satisfied.
In Figure 1, it is seen that the expression of S1PR1 and S1PR5 receptors in HUVEC and 143B cells is not statistically significant.
However, when the graphs are examined, it is seen that the mean values of S1PR1 expression in 143B cells decreased by approximately 5 times, while the mean values of S1PR5 expression increased by approximately 3 times. In addition, the standard error of mean values was quite small, as seen. These statistics need to be re-evaluated. The graphs should be reinterpreted accordingly.
The meaning of the single asterisk is stated in the explanation of Figure 1. However, the meaning of the double asterisk in the graph is not included in the explanation of Figure 1. The double asterisk should be explained.
The discussion section needs substantial improvement. You should start the discussion by listing key findings in your study. Discussion should start with listing the key results of the study, followed by a comment on how the results support the hypothesis.
Limitations should be stated and listed before the conclusion section.
Reference 33 mentioned in the reference section is not included in the manuscript. References should be reviewed again.
A 26% similarity was found in the manuscript by comparing the similarity report with a predetermined set of rules. The manuscript needs to be checked again to reduce the similarity rate.
Author Response
The following corrections/clarifications are required. The comments are being provided section-wise:
1. The introduction section is inadequate. The hypothesis is missing, and the aim is not well written. The structure of original articles should be satisfied.
Response: We have thoroughly revised and expanded the introduction by better clarify our hypothesis, provide a more comprehensive discussion of the clinical relevance of S1P, and strengthen the supporting rationale. We hope that the reviewer now finds the introduction clearer, more cohesive, and sufficiently detailed. Lines: 71-81 and lines: 82-94
2. In Figure 1, it is seen that the expression of S1PR1 and S1PR5 receptors in HUVEC and 143B cells is not statistically significant. However, when the graphs are examined, it is seen that the mean values of S1PR1 expression in 143B cells decreased by approximately 5 times, while the mean values of S1PR5 expression increased by approximately 3 times. In addition, the standard error of mean values was quite small, as seen. These statistics need to be re-evaluated. The graphs should be reinterpreted accordingly.
Response: The reviewer is right, an asterisk was missing in the graph regarding S1PR1; was have modified the text accordingly: “In contrast, relative to HUVECs, 143B cells display notably higher levels of S1PR2 and S1PR5 receptor expression and significantly lower expression of S1PR1.”. Regarding S1PR5, the rather low standard error suggests precision in the measurements, but probably the numerosity of the sample is not sufficient to obtain statistical significance.
3. The meaning of the single asterisk is stated in the explanation of Figure 1. However, the meaning of the double asterisk in the graph is not included in the explanation of Figure 1. The double asterisk should be explained.
Response: We thank the reviewer for noticing the error, we have now added the description of the double asterisk to the figure legend.
4. The discussion section needs substantial improvement. You should start the discussion by listing key findings in your study. Discussion should start with listing the key results of the study, followed by a comment on how the results support the hypothesis.
Response: We have added a new paragraph at the beginning of the discussion that summarizes the main findings of the manuscript and the supporting hypothesis, according to the reviewers’ suggestion. Lines: 267-273
5. Limitations should be stated and listed before the conclusion section.
Reponse: We have now added a paragraph with the limitations of the study before the conclusion section, as suggested by the reviewer. We hope that the reviewer now finds the discussion more linear. Lines: 341-358
6. Reference 33 mentioned in the reference section is not included in the manuscript. References should be reviewed again.
Reponse: The reviewer is right, the reference was missing in the text and has now been added in the introduction (now reference 4).
7. A 26% similarity was found in the manuscript by comparing the similarity report with a predetermined set of rules. The manuscript needs to be checked again to reduce the similarity rate.
Response: We have re-written and readapted a consistent part of the text, especially of the introduction and discussion, and are confident that this has reduced the similarity rate.

Reviewer 3 Report
Comments and Suggestions for Authors
Please, see the attachment

Author Response
The manuscript under review, entitled “Antagonizing the S1P-S1P3 axis as a promising anti-angiogenic strategy” is devoted to characterization of KRX-725-II and its derivatives, [Tic]4-KRX-725-II and [D-Tic]4- KRX-725-II pepducins by its action on proliferation, migration inhibition, tubulogenesis and sprouting. Interesting study with new data. I have some questions and commentaries that should be addressed before accepting the manuscript.
The thank the reviewer for the having carefully read our manuscript and for the comments, that have been addressed and hopefully improved the quality of the manuscript.
- The authors wrote that “We recently demonstrated that osteosarcoma (OS) cells secrete high S1P levels and that targeting the S1P-sphingomyelin pathway selectively impairs therapy resistant tumor subpopulations in acidic microenvironments” (lines 72-74). The same in the abstract (lanes 26-27). According to ref. 11, the authors used only three continuous cancer lines for that study. As far as I understand, they used no primary osteosarcoma cultures, isolated from patients with osteosarcoma, as well as performed no broad screening of osteosarcoma cancers. If this is true, then the above mentioned sentences are too generalized. The question is how often osteosarcoma cancers in patients secrete high S1P levels? Is this clinically relevant production? I suggest to the authors proved the claim with other studies in the literature or, otherwise, narrow the claim.
Response: We thank the reviewer for giving us the possibility of making this more clear. The cited study was performed on OS cell lines; in addition OS tissues collected from patients were used to demonstrate the relationship between acidosis and lipid accumulation in OS but also in other sarcomas and, more importantly, plasma from OS patients was used to assess the presence of S1P, both by ELISA and lipidomic analysis. Figure 3I of the cited study shows that S1P is significantly higher in patients before chemotherapy treatment and drops after chemotherapy, suggesting its clinical relevance as a prognostic marker. Also, plasma S1P levels have been found almost twice in ovarian cancer patients than in healthy controls (PMID: 15247129) and are prognostic in lung cancer (PMID: 23749868). S1P levels are also known to increase metastasis: PMID: 29269995, PMID: 28108260, PMID: 28052056, PMID: 22707406)
We have tried to substantiate the paragraph and added the mentioned references, as follows . (lines: 76-89): “Our prior work revealed that osteosarcoma (OS) 143B cells secrete high S1P levels, and targeting the S1P-sphingomyelin pathway selectively impairs thera-py-resistant, aggressive subpopulations within acidic tumor niche, and it reduces xeno-grafts tumor growth [12]. At the clinical level, serum S1P concentrations in OS patients decline following chemotherapy, suggesting a direct correlation between S1P and tumor progression [12]. These data, along with evidence from ovarian and lung cancers linking high S1P levels to metastatic spread [13-16], and poor prognosis [17][18], further highlight the therapeutic potential of S1P axis inhibition in OS. Targeting S1P-S1PR signaling could impair aggressive cancer cells while modulating angiogenesis, thereby suppressing vas-cularization and metastasis. However, the effect of S1PR inhibition on OS angiogenesis remains unexplored..”
- The authors used a concentration of 100 μM of pepducins to show anti-sprouting activity on OrganoPlate, however, it is very high concentration. They demonstrated statistically significant anti-tubulogenesis activity in a concentration of 10 μM. Why the authors chose such a high concentration of the compounds for anti-sprouting assay? Did they try lower concentrations, for instance, 10-20 μM? If they tried lower concentrations and saw no effects, it should be included in the results section.
Response: We tried different concentrations, including 10 mM, but we have seen a drug effect only a 100 mM. This is probably due to the difference between a traditional tubulogenesis assay, where the cells are seeded on top of the matrigel layer and do not form a real 3D structure, respect to the vessel formed in microfluidics. Here, the cells organize into a 3D vessel structure that more closely resembles physiological conditions. Indeed, physiological relevance of these systems could influence drug effectiveness, necessitating empirical determination of optimal dosing for each experimental setup, as observed in other cancer models (PMID: 25116894)
We have added this additional information in the result section and in the discussion. Lines: 233 and lines: 359-36, as follows: (This setup enabled passive perfusion, mimicking shear stress, and the formation of gra-dients of nutrients and their exchange within a 3D ECM (Matrigel). In these conditions, pepducins inhibited vessel sprouting only at 100 M. This discrepancy in respect to re-sults obtained with conventional tubulogenesis assays – where endothelial cells form simplified 2D networks on a Matrigel® layer - likely stems from the self-organization ability of endothelial cells to recapitulated 3D vascular complexity that is allowed and facilitated in such device. more closely mimicking physiological conditions. Such phys-iological fidelity can alter drug responsiveness, underscoring the need for empirical dose optimization in 3D systems, as observed in other cancer modelsX)
- The authors should discuss in the discussion section the issue of concentration. How clinically relevant could be these findings, if the pepducins show anti-angiogenic activity in such high concentrations? Is it possible to reach such concentrations of that compounds in blood stream? They should compare the obtained data with clinical trials of other S1P receptor antagonists in terms of concentrations – which concentrations of other S1P receptor antagonists were used in clinical trials?
Response: This is an interesting point. We have to point out that this is a preliminary study and that further investigation is needed to assess more physiological concentrations or perhaps additional modifications to the drug that might lead to a more specific targeting of tumor endothelial cells with less side effects (for example encapsulation, nanocarriers or chemical modifications). This said, the clinical trials that are currently present with S1PR inhibitors or modulators are mostly used for the treatment of multiple sclerosis or systemic lupus erythematosus and use a drug concentration that ranges from 0.5 mg daily to 20 mg daily (https://my.clevelandclinic.org/departments/neurological/depts/multiple-sclerosis/ms-approaches/sphingosine-1-phosphate-receptor-modulators?utm_source=chatgpt.com), concentrations that would probably be lower than the one employed in our assays. It's important to note that these dosages are specific to the conditions studied and the particular S1P receptor modulator used. Dosage regimens may vary based on factors such as disease severity, patient population, and therapeutic goals. Precise concentrations for the employed drugs should carefully be evaluated on the basis of the target, disease and the antagonized receptor.
We have added this information in the discussion. Lines: 341-349
- Pepducins are believed to be cell-penetrating peptides. How can it be that cell-penetrating peptides do not have cytotoxity in a concentration of 100 μM?
Response: This is another interesting point raised by the reviewer. We were actually quite surprised not to see an effect of the compounds on tumor cells. There are several possible reasons for the lack of phenotype. One possible reason is the different presence of intracellular lysosomes in tumor cells respect to endothelial cells. Indeed, it is known that a number of sarcomas develop a drug resistant mechanism that is dependent on high level of lysosome acidity, mediated by high expression of the vacuolar ATPase (V-ATPase). It has been shown that targeting the V-ATPase with the proton pump inhibitor omeprazole in a xenograft model of 143B OS cells sensitizes tumor cells to the conventional chemotherapy drug doxorubicin (PMID: 27566564), suggesting that 143B cells have high number of acidic lysosomes, responsible for drug ineffectiveness.
It is also known that lysosome acidity is further increase by extracellular acidosis, which develops with physiological tumor growth and can be assessed also in vitro (PMID: 33467731). Indeed, extracellular acidosis can further increase the number and acidity of lysosomes (PMID: 27566564); culturing 143B OS cells as spheroids, as we did for the migration and viability assays, can lead to the development of acidosis in the inner part of the spheroid, thus reducing drug effectiveness.
One last possible reason for the lack of phenotype in tumor cells is the presence of extracellular matrix. Indeed, when cultured as spheroids, the secretion of extracellular matrix can increase, and therefore act to limit diffusion and penetration of drugs, as seen in OS for doxorubicin (PMID: 36831562).
We have added this information in the discussion. Lines: 296-308
- The authors wrote “Since previous studies have shown that inhibiting the S1P pathway affects cell viability in 3D rather than in 2D culture [11], we further evaluated the effects on 143B cells grown in 3D using an indirect viability assay (Alamar Blue assay, Fig. 2D)”. However, I did not find in ref. 11 the comparison of cell viability between 3D and 2D culture. There are no even “Study of cell viability” or “Cytotoxity assay” titles in Methods section in that reference.
Response: Yes, the reviewer is right, we have extended a concept that we know for sure and give for granted, having worked for many years on the differences between 2D and 3D, the differences in IC50 of the drugs used in 2D vs 3D and the viability between the 2 conditions. For example, we have clearly assessed differences in IC50 in this paper: PMID: 36831562 that nevertheless does not take into account S1P signaling pathway. We also found a similar phenotype of reduced drug sensitivity from 2D vs 3D in chondrosarcoma spheroids (PMID: 29469166). This said, we are sorry that the concept is not formally assessed in the cited reference, where the difference shown between 2D and 3D refer mostly to lipid content. We have now narrowed down the claim: Lines: 127-128 “Since previous studies have shown consistent differences between 2D and 3D models in OS”
- Viability of OS spheroids is presented in unobvious way (Figure 2D). The most obvious way is a graph plotting the percentage of cell viability versus concentration. There are approaches to study viability of cells in spheroids, that allow to calculate the percentage of cell viability versus concentration. For instance, [1].
Response: The referenced paper uses a dual viability stain using fluorescein diacetate (FDA) and propidium iodide (PI) in combination with flow cytometry. We agree that this method is more precise and direct respect to the one we have used (Alamar blue staining), which is characterized by the conversion of resazurin to resorufin, thus assessing the metabolic activation of the cells. We will take into account the suggestion of the reviewer and for future studies we will start using double cell staining to better assess viability.
7) The most interesting results were obtained by using OrganoPlate. But, in my subjective opinion, the images in Figure 5C are not quite illustrative. Did the authors try other incubation time, for instance, 48 h? If yes, maybe they can add some additional images in the supporting material.
Response: This experiment was performed in the following way: GFP-positive HUVEC cells were seeded in the upper channel of the microfluidic chip, wherease the matrix gel in the adjacent and lower channel. A after 24 hrs, vessel formation against the hydrogel wall was assessed by GFP fluorescence imaging using confocal microscopy. At this time point, an angiogenic cocktail was added to the lower channel of the microfluidic chip and pepducins to the upper channel. Imaging acquisition was performed 24 hrs after addition of the drugs. Live cell imaging (using GFP fluorescence in HUVEC cells) was performed 24 hours post-drug. Extending the timepoint beyond 24 hrs (e.g., to 48 hrs), resulted in HUVEC cells spreading on the upper and lower surfaces of the Matrigel channel, leading to aberrant sprouting outside the gel matrix. Thus, the 24-hrs time point was selected because sprouting cells remained embedded within the Matrigel, avoiding surface migration and cells were not seen to crawl on top or on the bottom of the channel. Post-24-hour imaging was excluded as results became non-representative. This protocol is detailed in the Methods section (paragraph 4.7), summarized in Figure 5A’s legend, and contextualized in the Results (line 221).
8) Migration assay is usually performed in other way, by using transwells with 8 μm pores, when the cells migrate from apical to basolateral surface of the membrane. It is the way, how the authors studied cell migration in their previous paper (ref.11). Why here they used other approach? What they wanted to study? Figure 2E is quite confusing. What the authors wanted to show on these images? There is difference between control and compounds (especially with [Tic]4-KRX-725-II) on 2E, however, there is no difference at all on the diagram 2F.
Response: Conventional migration assays, such as transwell assays, require cell tripsinization to enable migration through membrane pores. This disrupts cell-to-cell contact, a process inconsistent with the cohesive cell behavior observed in tumors. In contrast, our spheroid-based assay preserves intact 3D cell clusters. Spheroid attachment to the plate surface mimics physiological tumor cell detachment and migration through extracellular matrices. Migration is quantified by measuring the increase in spheroid spread area over time. Specifically, migration area was calculated as the difference between the spheroid spread area at 24 hours and the initial area at 0 hours (when spheroids were first attached to the plate; Fig. 2E). The reviewer has seen right, it seems that there is a trend of increased migration, although this effect was not significant. We changed the text of the manuscript as follows (at lines 125-129): “. Migration was quantified as the difference in spheroid spread area between the initial time point (0h, black circles), and 24 hrs post-treatment (red circles), a metric reflect-ing invasive potential akin to in vivo tumor behavior . Unexpectedly, drug-treated sphe-roids showed a trend toward increased migration, though no treatment significantly impaired migration compared to controls ”
Minor points:
- Did the authors check the integrity of extracted tRNA? They are suggested to include agarose gel electrophoresis of the extracted samples of tRNA in the supporting material.
Response: We are sorry but do not have gels that assess the integrity of mRNAs. We do check RNA purity with an OD 260/280 ratio 1.8-2.0 and an OD 260/230 2.0-2.2.
2) Did the authors check the efficacy of used primer pairs during qPCR? Did they design the primer pairs or used previously published ones? Please, add this information to the Methods section. Please, provide in the supporting material an agarose gel electrophoresis of qPCR amplicons, demonstrating high specificity of all primer pairs and lack of additional products.
Response: The primers were designed by us; we have implemented the methods section with the description of how we designed primers (paragraph 4.2). We normally do not run qPCR amplicons on agarose gel, but the Syber Green methods has the advantage of showing melting curves at the end of the run. From the melting curve it is possible to check whether primers give rise to one or more PCR products.
Please find here attached (file) the melting curves of the 5 different primers. We hope that this answers to the reviewers’ request.
3) Please, add full name of FABP4 (lane 59).
Response: We have corrected the name, according to the reviewers request.
4) Please, add full name of SphK (lane 70).
Response: We have corrected the name, according to the reviewers request.
[1] Vej-Nielsen, J.M., Rogowska-Wrzesinska, A. (2021). 3D-ViaFlow: A Quantitative Viability Assay for Multicellular Spheroids. In: Brevini, T.A., Fazeli, A., Turksen, K. (eds) Next Generation Culture Platforms for Reliable In Vitro Models. Methods in Molecular Biology, vol 2273. Humana, New York, NY.

Round 2
Reviewer 2 Report
Comments and Suggestions for Authors
I would like to thank the authors for providing a very detailed and comprehensive response to my comments from the previous review.
Reviewer 3 Report
Comments and Suggestions for Authors
Now I believe that the manuscript has been significantly improved and now can be accepted for the publication in Metabolites. Figures 1, 2, 3 and 5 do not have clear white background.